Spawning aggregation behavior and reproductive ecology of the giant bumphead parrotfish, Bolbometopon muricatum, in a remote marine reserve

Muñoz Roldan C. 1 roldan.munoz@noaa.gov
Zgliczynski Brian J. 2
Teer Bradford Z. 1
Laughlin Joseph L. 3
1 National Marine Fisheries Service, Beaufort Laboratory, National Oceanic & Atmospheric Administration , Beaufort, NC , USA
2 Center for Marine Biodiversity & Conservation, Scripps Institution of Oceanography , La Jolla, CA , USA
3 Mariculture Hawaii LLC , Ashland, OR , USA
Toonen Robert
Electronic publication date: 2014 Nov 25
Publication date: 2014
Volume: 2
Electronic Location ID: e681
Received 2014 Sep 11; Accepted 2014 Nov 9
Copyright: © 2014 Muñoz et al.
Copyright year: 2014
Copyright holder: Muñoz et al.
License: This is an open access article, free of all copyright, made available under the Creative Commons Public Domain Dedication. This work may be freely reproduced, distributed, transmitted, modified, built upon, or otherwise used by anyone for any lawful purpose.
License URL: https://creativecommons.org/publicdomain/zero/1.0/

Keywords: MPA, Scarinae, Conservation, Mating system, Lek, Alternative reproductive behavior, Threatened species, Site fidelity, Indo-Pacific, Atoll

Funding: NOAA Proactive Species Conservation Program National Marine Fisheries Service Southeast Fisheries Science Center Support was provided by the NOAA Proactive Species Conservation Program to RCM and BJZ. Support for preparation of the manuscript was provided by the National Marine Fisheries Service Southeast Fisheries Science Center to RCM. The funders had no role in study design, data collection and analysis, decision to publish, or preparation of the manuscript.

==============================
The giant bumphead parrotfish (Bolbometopon muricatum) has experienced precipitous population declines throughout its range due to its importance as a highly-prized fishery target and cultural resource. Because of its diet, Bolbometopon may serve as a keystone species on Indo-Pacific coral reefs, yet comprehensive descriptions of its reproductive ecology do not exist. We used a variety of underwater visual census (UVC) methods to study an intact population of Bolbometopon at Wake Atoll, a remote and protected coral atoll in the west Pacific. Key observations include spawning activities in the morning around the full and last quarter moon, with possible spawning extending to the new moon. We observed peaks in aggregation size just prior to and following the full and last quarter moon, respectively, and observed a distinct break in spawning at the site that persisted for four days; individuals returned to the aggregation site one day prior to the last quarter moon and resumed spawning the following day. The mating system was lek-based, characterized by early male arrival at the spawning site followed by vigorous defense (including head-butting between large males) of small territories. These territories were apparently used to attract females that arrived later in large schools, causing substantial changes in the sex ratio on the aggregation site at any given time during the morning spawning period. Aggression between males and courtship of females led to pair spawning within the upper water column. Mating interference was not witnessed but we noted instances suggesting that sperm competition might occur. Densities of Bolbometopon on the aggregation site averaged 10.07(±3.24 SE) fish per hectare (ha) with maximum densities of 51.5 fish per ha. By comparing our observations to the results of biennial surveys conducted by the National Oceanic and Atmospheric Administration (NOAA) Coral Reef Ecosystem Division (CRED), we confirmed spatial consistency of the aggregation across years as well as a temporal break in spawning activity and aggregation that occurred during the lunar phase. We estimated the area encompassed by the spawning aggregation to be 0.72 ha, suggesting that spawning site closures and temporal closures centered around the full to the new moon might form one component of a management and conservation plan for this species. Our study of the mating system and spawning aggregation behavior of Bolbometopon from the protected, relatively pristine population at Wake Atoll provides crucial baselines of population density, sex ratio composition, and productivity of a spawning aggregation site from an oceanic atoll. Such information is key for conservation efforts and provides a basic platform for the design of marine protected areas for this threatened iconic coral reef fish, as well as for species with similar ecological and life history characteristics.

Introduction

Human impacts to terrestrial and marine communities are widespread and typically begin with the local extirpation of large-bodied animals (Pandolfi et al., 2003; Morrison et al., 2007). Large-bodied species play an important ecological role and many are key species that may be critical for maintaining long-term ecosystem stability (Bellwood, Hoey & Choat, 2003; Sadovy et al., 2003; Pandolfi et al., 2005). Large predators can shape the number, distribution, and behavior of their prey, while large herbivores can act as ecological engineers by shaping the structure and species composition of plant or algal communities (Morrison et al., 2007; McCauley et al., 2010). The challenge of understanding how large-bodied species contribute to ecosystem structure and community organization is often confounded by their rarity. The giant bumphead parrotfish (Bolbometopon muricatum, A. Valenciennes, 1840) illustrates this challenge.

Bolbometopon muricatum (Labridae, Scarinae; Westneat & Alfaro, 2005) is a monotypic genus (hereafter Bolbometopon) and the largest herbivorous and corallivorous fish on coral reefs, reaching 150 cm total length (TL) and over 75 kg total weight (Gladstone, 1986). It is slow-growing and long-lived, with delayed reproduction and low replenishment rates (Choat & Robertson, 2002; Hamilton, Adams & Choat, 2008; Bellwood & Choat, 2011; Taylor et al., 2014). Even moderate levels of exploitation have led to severe declines in size-structure and abundance throughout much of its range (Dalzell, Adams & Polunin, 1996; Aswani & Hamilton, 2004; Donaldson & Dulvy, 2004; Dulvy & Polunin, 2004; Olds et al., 2014). This is particularly evident where it is a highly prized fishery target and an important cultural resource (Sadovy, 2005). The most detrimental factors contributing to its vulnerability are that Bolbometopon exhibits predictable behavior and is gregarious, often sleeping and feeding in large groups in shallow water (1–15 m deep), making it highly susceptible to exploitation by night spearfishing and netting of daytime feeding schools (Johannes, 1981; Hamilton & Choat, 2012). For example, night spearfishing increased with the advent of underwater flashlights in the 1970’s, and in the western Solomon Islands led to overexploitation and the disappearance of sleeping aggregations that had persisted and supported subsistence fishing for generations (Hamilton, 2004). In response to declining populations throughout its range, Bolbometopon was listed as Vulnerable in 2007 by the International Union for Conservation of Nature (IUCN) and is considered a Species of Concern by the U.S. National Marine Fisheries Service (NMFS). Overfishing has apparently altered the behavior of Bolbometopon and led to a general avoidance of humans throughout much of its range; it is known as the wariest of parrotfishes and in most locations individuals are difficult to approach underwater (Myers, 1999).

Australia’s Great Barrier Reef (GBR), however, has no commercial fisheries for parrotfishes. As such, these reefs support healthy populations of Bolbometopon where schools of 30–50 individuals can be observed regularly (Bellwood, Hoey & Choat, 2003; Bellwood & Choat, 2011). On the GBR, Bolbometopon play an important ecological role where individuals are capable of bio-eroding over 5 tons of reef carbonate each year (Bellwood, Hoey & Choat, 2003). Because of its large size, feeding rates, and schooling behavior, Bolbometopon may hold a keystone role as a major coral consumer, algal grazer, and bio-eroder on coral reefs (Bellwood, Hoey & Hughes, 2012; McCauley et al., 2014). In overfished locations, negative effects may include significant disruption to coral community structure, reductions in reef structural stability via invasive erosion by echinoids, and dramatic reductions in sediment transport (Bellwood, Hoey & Choat, 2003; but see McCauley et al., 2014). Given Bolbometopon’s vulnerability to overexploitation, ecological role, and general avoidance of humans, comparative studies of its biology and ecology from additional unexploited populations are urgently needed to provide critical insights for the development of recovery and management plans (Rowe & Hutchings, 2003; Comeros-Raynal et al., 2012). The only detailed published spawning event comes from the GBR and consists of a single observation made nearly thirty years ago (Gladstone, 1986); minor notes of individuals believed to be in reproductive groups also exist (Johannes, 1981; Choat & Randall, 1986; Daw, 2004; Hamilton, 2004), and additional spawning observations have recently been reported from elsewhere (Hamilton & Choat, 2012; Wu, 2013).

Indirect studies, inferring periods of seasonal reproduction from fishing and indigenous ecological knowledge have produced conflicting results: reports of spawning behavior or ripe gonads from fish have been observed or collected throughout lunar and circadian cycles (Johannes, 1981; Aswani & Hamilton, 2004; Hamilton, Adams & Choat, 2008). Hamilton, Adams & Choat (2008) examined the gonads of 103 Bolbometopon collected during artisanal spearfishing trips in the Solomon Islands and concluded that Bolbometopon appeared to be functionally gonochoristic (non-sex changing), similar to only two other parrotfishes (Robertson, Reinboth & Bruce, 1982; de Girolamo, Scaggiante & Rasotto, 1999). However, Hamilton, Adams & Choat (2008) found males in all three sampled populations were larger than females, suggesting an advantage to being male when large, and consistent with a protogynous (female to male) pattern of sex change. They also found evidence that some intermediate sized males appeared to invest as heavily in gonad development as ripe females, indicative of sperm competition. Further conclusions were not possible without behavioral observations of spawning, and it is currently unknown whether or not sex-change occurs in any location across the geographical range (Hamilton, Adams & Choat, 2008; Hamilton & Choat, 2012).

Direct in situ studies of reproduction and spawning in Bolbometopon are scarce. Gladstone (1986) observed a single spawning event on the GBR that took place approximately 1.5 h after sunrise on an outgoing tide in 15 m water depth. The spawning event involved a pair of Bolbometopon breaking off from the top of a tightly packed school of approximately 100 individuals. The pair then slowly ascended to one meter below the water’s surface, releasing a clearly visible gamete cloud, followed by a slow descent back to the school. Muñoz et al. (2012) first documented ritualized head-butting behavior of Bolbometopon associated with early-morning spawning events of an unexploited and intact population at Wake Atoll, and suggested that sites where Bolbometopon are abundant may give rise to intense competition (Roberts & Polunin, 1991) and sexual selection. This recent documentation of such dramatic social and reproductive behavior from a large and iconic species emphasizes that much remains to be studied concerning the biology and ecology of Bolbometopon to achieve effective management and conservation (Olds et al., 2014). For example, Hamilton & Choat (2012) indicated that no evidence exists that Bolbometopon feeding schools merge to achieve a larger mass of reproductive individuals, or to provide a basis for spawning closures as a management option. Although no-take reserves were suggested as the best option for longer term protection of Bolbometopon, Hamilton & Choat (2012) proposed that reserves would need to encompass a minimum area of 6 km2. These questions indicate the need for behavioral field studies to acquire basic information regarding the reproductive ecology of Bolbometopon, including spawning site location, frequency of utilization, and spatial extent of reproductive individuals.

Here we provide much needed information pertaining to the reproductive ecology of Bolbometopon from the protected and intact population of Wake Atoll. Our goal is to improve upon the key single published spawning observation made on the GBR nearly thirty years ago (Gladstone, 1986). This contribution can assist others in identifying reproductive behavior in other locations and identifying and protecting additional spawning sites, key steps in conserving this unique and threatened member of Indo-Pacific coral reefs.

Materials and Methods

Permission to conduct field research at Wake Island was granted by Euretha T. Dotson, United States Air Force, Wake Island, HI. All research was conducted in accordance with the Animal Welfare Act (AWA) and with the U.S. Government Principles for the Utilization and Care of Vertebrate Animals Used in Testing, Research, and Training (USGP) OSTP CFR May 20, 1985, Vol. 50, No. 97. The study was conducted on free-living wild animals in their natural habitat and solely involved observations of animals and noninvasive measurements.

Wake Atoll (19°18′N, 166°37′E) is a U.S. Pacific Remote Island, National Wildlife Refuge, and Marine National Monument co-managed by the National Oceanic and Atmospheric Administration (NOAA) and the U.S. Fish and Wildlife Service (USFWS), but with access controlled (since 1934) by the U.S. Department of Defense (DOD). Because of its remote location and its control by DOD, commercial fishing in waters surrounding Wake Atoll has been excluded and all recreational fishing for Bolbometopon is prohibited. As such, populations can be considered pristine (island-wide mean of 2.97 ± 0.96 SE individuals per ha, Lobel & Lobel, 2004; Zgliczynski et al., 2013).

We used a variety of underwater visual census methods (UVC) to document Bolbometopon populations around Wake Atoll. These included snorkel, scuba, and towed-diver surveys. Towed-diver surveys are useful for sampling large-bodied fishes that are relatively rare and have comparatively large home ranges (Richards et al., 2011), although they may underestimate species that engage in avoidance behavior when approached by a motorized tow-vessel (Rizzari, Frisch & Connolly, 2014). In total, the team completed over 100 h of UVC surveys from 12 to 25 August 2011. Underwater visibility ranged from 4.5 m to >30 m, depending on the tidal state (ebb tide drained the atoll lagoon, decreasing visibility). General underwater conditions can be found in Lobel & Lobel (2008). We selected study sites along the outer fore reef using the results of biennial towed-diver surveys conducted by the NOAA Coral Reef Ecosystem Division (CRED) (see Fig. 3 in Muñoz et al., 2012). Detailed CRED survey methods can be found in Richards et al. (2011). These surveys identified spatial consistency of higher Bolbometopon densities at the southwest (SW) side of the island across years, suggesting that Bolbometopon may form true spawning aggregations at Wake Atoll (sensu Domeier, 2012). Our surveys initially targeted this SW location, although we eventually also surveyed the west, much of the south, and a smaller area on the north of the atoll (Fig. 1). Prevailing winds from the northeast and the location of the small boat harbor in the south central portion of the atoll made it difficult to sample the northeast, east, and southeast areas.

Figure 1 Location of surveys for Bolbometopon muricatum with scuba, snorkel, and towed-divers conducted between 12 and 25 August 2011 at Wake Atoll.

Location of surveys is shown in red.

During the study, sunrise and sunset occurred from 0633–0637 h, and 1915–1924 h, respectively. Our observations took place from 0624 to 1926 h and surveys were geographically logged with a hand-held global positioning system (GPS) (Nanami & Yamada, 2008; Colin, 2012a) to estimate area surveyed and the location of significant behavioral events (e.g., spawning, aggression). These GPS units calculate the area surveyed based on the external edge of the total number of track points that are automatically recorded during an observation period. We conducted spawning observations from shortly before sunrise until the cessation of reproductive behavior, which we defined as the point when the majority of individuals commenced feeding and dispersed from the spawning site (Muñoz et al., 2010). We recorded reproductive and spawning behavior of Bolbometopon using high definition video (Canon Vixia HF S200, Sony HDR-HC9), and digital still photography (Nikon D300, Canon G9) while slowly swimming haphazardly throughout the SW location.

Video analyses

For each day of our study, we analyzed the first video/dive of the day (corresponding with the morning spawning period) from the SW location to estimate the number of fish in the aggregation area with a mean count method (Conn, 2011; Schobernd, Bacheler & Conn, 2014) and CyberLink PowerDirector 10 software. On two days, defective first videos made it necessary that we analyze the second video/dive of the day. Each video was approximately 70 min in length. At two-minute increments we quantified the number of fish in a video frame and calculated the mean for each video. If blurry conditions impeded counts at the two-minute mark, we advanced or reversed frame by frame until all individuals visible at the two minute mark could be counted (Kenyon et al., 2006). We also used these videos to classify sex/social phase, to estimate sex ratios of the aggregation, and to classify and count overt behaviors and morphological (color, erection of fins) displays that were apparent during the morning spawning period. All distinct behaviors and morphological displays that we observed sequentially were counted. On occasion, a large school of fish moved through our field of observation resulting in simultaneous behavior or displays from many individuals. To minimize video processing time we recorded these simultaneous interactions only once. The behavioral data represent a combination of sequential records of individual behavior together with simultaneous behaviors recorded from groups, so for analyses we normalized the behavioral observations for each day by dividing by the mean number of fish estimated from the video for that day.

Based on the work of Hamilton, Adams & Choat (2008) we classified sex with a combination of dimorphic morphology and behavior in the field centered on the fact that male Bolbometopon are typically larger than females. Although all adults appear to develop a cephalic hump on the forehead (Liu & de Mitcheson, 2011), we assigned sex as female based on smaller body size and distinct forehead morphology. In females the forehead slopes backward from the beak while male fish have foreheads that tend to rise vertically from the beak (Fig. 2A, also see Muñoz et al., 2012). In the field, behavior was also used to differentiate females from males. We assumed that most morphological female fish interacting with morphological and behavioral males were indeed female, though certain populations of Bolbometopon show overlap in size distribution of males with females (Hamilton, Adams & Choat, 2008). In those cases, some morphological female fish may be males that have not yet developed the male forehead morphology. In our study, all observations of courtship and spawning that we observed were between dimorphic fish, presumably female and male (Warner & Robertson, 1978; Suzuki et al., 2008; Muñoz et al., 2012).

Figure 2 Reproductive ecology of Bolbometopon muricatum at Wake Atoll.

(A) Typical coloration and morphology of male and female Bolbometopon. (B) Four male Bolbometopon (denoted with arrows) stationed on their territories. Note male on right high in water column at apex of ascent, as well as full face blanching displayed by all four males. Also see Video S2. (C) Male Bolbometopon on territory stationed above a school of females. We frequently encountered male Bolbometopon at the spawning aggregation site high in the water column, and males often finished courtship interactions with a slow ascent into the water column. (D) Detail of courtship looping behavior illustrating temporary dichromatism between male and female fish. See Video S3. (E) Pale diffuse bar often displayed by male Bolbometopon, indicated with white arrows on five fish. (F) Parallel swim assessment behavior between two stationary male Bolbometopon. Note male in foreground displaying full blanched coloration. See Video S4. (G & H) Pair spawning sequence of Bolbometopon. Note the temporary monochromatism (including fully blanched face in both male and female) preceding gamete release. Panels (G) and (H) taken one second apart. Also see Video S5.

Data analyses

Analyses were conducted with SigmaPlot 11.0. All data were tested for normality with the Kolmogorov–Smirnov test and for homoscedasticity with Levene’s mean test. Since the data consistently violated parametric assumptions we employed nonparametric statistical methods. We used Mann–Whitney Rank Sum tests to compare the number of Bolbometopon at the aggregation site in relation to moon phase, to compare the social phases of Bolbometopon present at the aggregation site, and to compare the density of Bolbometopon observed in relation to aggregation days versus non-aggregation days. We counted all overt behaviors observed on videos and used a Kruskal-Wallis one-way ANOVA on ranks with a Tukey test to compare the mean number of different behaviors observed during the morning spawning periods. We used a Spearman rank correlation to examine the relationship between mean number of Bolbometopon at the aggregation site and frequency of behaviors observed per morning observation. To estimate the spatial extent of the aggregation site, we determined the area encompassed by all the GPS tracks recorded during observations where we observed Bolbometopon assembled for spawning, including locations where we observed spawning itself. We combined this estimate of area surveyed while Bolbometopon were present at the aggregation site with the mean number of fish counted on videos of the aggregation site each day to estimate the mean density of Bolbometopon present on aggregation days. We used GPS track logs to estimate area surveyed and Bolbometopon densities on non-aggregation days. We also used data from towed-diver surveys conducted by NOAA CRED from 2005 through 2014 (a total of five survey years including 2005, 2007, 2009, 2011, 2014) to estimate the densities of Bolbometopon at the aggregation site compared with other sites around Wake Atoll. We employed Mann–Whitney tests with the CRED data to examine the relationship between densities of Bolbometopon at the aggregation site versus other sites during morning surveys, and compared these data with afternoon surveys.

Results

General observations of Bolbometopon mating system

Our observations of Bolbometopon reproductive activity took place from 0624 to 0935 h. Within the first minutes of initiating the study it was clear that the SW location would be a suitable site to study the aggregation of Bolbometopon. For example, on 13 August 2011 we entered the water at 0630 h (near sunrise) and found ourselves floating directly above a relatively stationary, tightly packed school of approximately 50–75 Bolbometopon within a 4–5 m deep coral groove of the reef crest. This Bolbometopon school may have been a nighttime sleeping aggregation, observed under similar conditions (sunrise) and with a similar composition as Gladstone (1986), consisting of female and male fish that slowly dispersed over the course of 30–45 min.

The result of this dispersion was that we observed females leaving the sleeping aggregation while reassembling into smaller reproductive schools of various size (10s to 100s of individuals, Video S1). These reproductive schools were distinct from feeding schools in that they consisted mostly of female fish (85.5 ± 3.79% female, n = 6 schools) that swam slowly around the general aggregation area. On one occasion, however, we observed a school that contained 87.5% male fish (21 males, 3 females).

In contrast to the slowly moving schools of female fish, large males stationed themselves along the benthos, separated approximately equidistantly 4–6 m apart from one another. Every few minutes in an asynchronous fashion, these males would leave their station on the bottom and slowly, with all fins generally erect and with slow exaggerated movements of the caudal fin, rise up from their benthic station to within 0.5–1 m of the surface. The males would hover in the water column for several minutes before undertaking a slow descent to their benthic station, usually maintaining erect fins and exaggerated movements of the caudal fin. The asynchronous ascents and descents coupled with slow, exaggerated movements appeared to be long-distance displays or advertisements by the males, and this behavior took place throughout the morning spawning period (Figs. 2B and 2C, Video S2). As the reproductive schools of females swam over the aggregation area they were intercepted and vigorously displayed to by the stationary males (Fig. 2D, Video S3), often less than 1 m directly in front of the female’s face, with a modified looping behavior (a well-known labroid courtship behavior, Reinboth, 1973; Moyer & Yogo, 1982; Colin & Bell, 1991). A courting male would frequently finish the particular courtship interaction with a slow ascent into the water column above the solitary female or school of females (Fig. 2C). We believe that territory establishment by males and courtship of either individuals or groups of females, together with additional behaviors described below illustrate a lek-like mating system (Moyer & Yogo, 1982) for Bolbometopon at Wake Atoll.

Most of the morning spawning period was associated with temporary dichromatism between male and female fish that could be rapidly displayed or hidden (Fig. 2D, Video S3, also see Hamilton & Choat, 2012). Courting males’ foreheads and entire faces extending posterior to the eyes would blanch, along with the lower posterior portion of the body and extending onto the caudal fin. White vertical bars along the flanks completed the looping male’s temporary dichromatism. Females usually responded to looping by displaying a series of (4–5) pale bars along the back and flanks, similar to those displayed by the male (Fig. 2D, Video S3). The bars displayed by the female would immediately fade following the male’s departure, while the bars and blanching along the body and caudal fin would similarly fade in males, with blanching of the face the last to fade following an interaction. However, return to normal male coloration (see Fig. 2A) was dependent on whether the male then turned its attention to another female or adjacent male, in which case the blanched face would be retained and courtship or aggressive interactions might follow.

Blanching of the face, as well as a single, sometimes diffuse, wide bar midway along the flank were common temporary color patterns of male Bolbometopon (Fig. 2E). We rarely saw females displaying the single wide bar (except seconds before spawning), but could often pick out males in a mixed-sex school by their display of the single bar upon the approach to the school of another male. Blanching of a male’s face, ranging from partial blanching anterior to the eye to extensive blanching extending posterior to the eye, was another common temporary color pattern. Temporary colors and the extent of their display may function as communication signals, with more extensive blanching apparently used to produce a stronger signal (Fig. S1). For example, extensive blanching of the face was often associated with more vigorous interactions such as looping (Fig. 2D), water column displays (ascents/descents) by males (Figs. 2B and 2C), or aggressive interactions such as male–male chases (Fig. S1) or parallel swims (Fig. 2F).

We most often observed aggressive interactions such as chases between two males (rarely between a male instigator and female recipient), and these consisted of an instigator that used the caudal fin to rapidly pursue a recipient, causing the recipient to flee the area, also with the caudal fin (in contrast to typical labroid sculling with pectoral fins). Given that the recipient in a chase fled the area, these interactions were likely between individuals that were mismatched in competitive ability or their value of a particular resource (e.g., territory) in the vicinity (Krebs & Davies, 1993). Chases could also be accompanied by blanching of the posterior body and caudal fin of the instigator, as well as the barring also seen in courtship (Fig. S1). Recipients of a chase responded with a diffuse wide bar, normal (unblanched) face, or partially blanched face (Fig. S1, Video S4). Another aggressive interaction that appeared to involve an assessment of competitive ability or resource value was a synchronous parallel swim between two males. Parallel swims were defined as two males that were swimming at approximately the same rate in the same direction and were either each displaying with a diffuse white bar, blanched face, or in more vigorous parallel swims, with full blanched coloration as seen in courtship (Fig. 2F, Video S4). Parallel swims ended when one or both members of the interaction reversed direction, or changed color or orientation such that the swimming and coloration were no longer in synchrony. Sometimes chases led to parallel swims, which presumably occurred when the recipient and instigator were more closely matched in competitive ability (Clutton-Brock & Albon, 1979). Head-butting between two males was the culmination of aggressive interactions, apparently related to the establishment of territories prior to spawning. In these cases, contestants were presumably closely matched in competitive ability or each greatly valued the contested resource such that the benefit to be gained by head-butting outweighed the costs of these violent interactions (Muñoz et al., 2012).

We rarely observed overt aggressive interactions between females, but did note the occasional approach of one female by another with an open beak, which usually displaced the recipient. Females engaged in these interactions often displayed a fully blanched face. Another behavior observed from females was a sustained (∼10 s) lean perpendicular to the instigator’s anterior-posterior body axis, accompanied by no obvious color change. These behaviors were not observed with sufficient frequency to be quantified.

Spawning occurred in the early morning following variable periods of movements by reproductive schools of females, male–male aggression, and courtship by males. Due to low light, spatial arrangement of stationary males, and low water clarity conditions related to the outgoing tide (see below), only 17 spawning events were clearly observed (Figs. 2G and 2H, Video S5). Gamete release occurred high in the water column (approximately 1–2 m below the surface) with spawning fish sometimes breaching the surface. We only observed spawning between pairs of male and female fish and would not characterize reproduction in Bolbometopon as including a spawning “rush” with gamete release at the apex. Rather, when apparently ready to spawn, an individual female with a fully blanched face would slowly approach, sometimes at an angle ≤ 45°, a stationary male hovering high in the water column. Responding with erect fins and also displaying a fully blanched face (temporary monochromatism, Moyer & Yogo, 1982), the male then moved into position alongside the female and the pair swam leisurely together horizontally, touching flanks for approximately five seconds followed by visible but faint gamete release, with the smaller female positioned slightly behind the larger male, with her forehead approximately in line with the male’s eye. The pair then slowly parted, with the female descending to the benthos and the male either remaining in the water column or also descending, but separate from the female. On four occasions we noticed males delivering a post-spawning loop display (Tribble, 1982) after the pair had separated following gamete release. We never observed successful interference spawning (streaking) of a spawning pair by either males or females, although in three instances we observed behavior that suggested interference spawning and sperm competition may sometimes take place. For example, in one instance a spawning pair was initially rapidly approached by another male who later slowed its rate of swimming and turned away before it reached the location of gamete release, as if it had arrived too late to interfere. On another occasion we observed a male in the water column twice loop a female that had approached with a fully blanched face and 45° orientation as if ready to spawn, but subsequently chase the female briefly in the opposite direction from its approach. When we reviewed this segment on video we noticed a large male nearby to the approaching female, so the looping male may have been chasing the female away from the adjacent male in response to a perceived threat of spawning interference.

Quantitative characteristics of Bolbometopon spawning aggregations

Lek mating systems are a form of polygynous reproduction that feature male defense of territories that appear too small to offer useful resources to visiting females (Fig. 2B and Fig. S2, Video S2). Males congregate on these communal display areas for the sole purpose of attracting and courting females who visit lek sites for mating (Wilson, 1975; Emlen & Oring, 1977). Our observations of Bolbometopon from the general aggregation area over multiple days detected the large schools of females that arrived each morning during the spawning period to mate with the males stationed on their territories. These reproductive schools of females visiting the aggregation area repeatedly altered the proportion of males present over the course of our observations. Fig. 3 shows three representative days on the aggregation area where the proportion of males during the morning spawning period changed from 100% male (stationed on individual territories awaiting females) to 0.89–40% male as schools of females arrived in the area and were courted by males, finally increasing as the reproductive schools moved away from the vicinity.

Figure 3 Large changes in sex ratio characterize the spawning period of Bolbometopon muricatum as reproductive schools of females move over the aggregation site at Wake Atoll.

Three representative days are illustrated from the morning spawning period at the SW aggregation site. Numbers below points indicate examples of numbers of fish counted from videos each two minute sampling period.

Our observations at the Bolbometopon aggregation site started on 12 August 2011, one day before the full moon, and continued until 25 August 2011, three days before the new moon. We observed large aggregations of Bolbometopon assembled for spawning around the full and last quarter moon, with peaks in aggregation size apparent just prior and just following the full and last quarter moon, respectively (Fig. 4). During those fourteen days, we observed a distinct break in spawning at the aggregation site that persisted for four days. We only observed a single male Bolbometopon at the aggregation site from 16 to 19 August, 2011, despite conducting extensive surveys throughout the day to most of the accessible (due to prevailing winds and logistics) portions of the atoll (Figs. 1 and 4). Bolbometopon had all but disappeared from vast regions of Wake Atoll from 16 to 19 August, and surveys that totaled nearly 360 ha (500 times greater survey area than that utilized [0.72 ha] to observe fish assembled at the aggregation site) were not successful in locating more than ten individuals, mostly singletons. On 20 August Bolbometopon began returning to the aggregation site and we observed them spawning the following day (Fig. 4). We observed a greater number of Bolbometopon at the aggregation site around the last quarter moon compared to the full moon (mean ± SE = 8.18 ± 3.75 vs. 6.09 ± 2.89 fish per morning observation, respectively), but the difference was not significant (Mann–Whitney test: T = 19.0, P = 0.91, nfull = 4 days, nlast 1/4 = 5 days). Over all days that fish were assembled at the aggregation site, we observed significantly more females (4.12 ± 0.61) than males (2.27 ± 0.61) per two-minute sampling interval (Mann–Whitney test: T = 73246.0, P < 0.001, n = 283 sampling intervals over 9 days).

Figure 4 Changes in aggregation size estimates of Bolbometopon muricatum and area surveyed at Wake Atoll.

Each bar shows the mean from two minute sampling intervals of a single approximately 70 min video recorded during the morning spawning period from the SW aggregation site. Shaded bars indicate days where we observed spawning. No spawning was observed on 22 August but large numbers of fish were present and we did hear the impact from a head-butt. It is likely that spawning occurred on 22 August, because in all but one case we observed or heard head-butts on days where we also observed spawning. We observed a single male fish on 16 August at the aggregation site but the sighting did not coincide with the two minute sampling intervals for that day. Filled black circles indicate survey effort in ha. Our survey effort on aggregation days was restricted to the aggregation site (approximately 0.72 ha), whereas on non-aggregation days we expended considerable effort surveying around the atoll in search of Bolbometopon.

We classified overt behaviors and distinct color patterns from videos recorded during the nine mornings that Bolbometopon were assembled at the spawning aggregation site. Since overt interactions between females were rare, we recorded the frequency of the following behaviors: head-butt between males (either witnessed or heard), chase between males, parallel swim between males, looping/courtship between males–females, and ascents/descents (males). We also recorded the number of distinct changes in coloration that we observed separately from one of the previous behaviors, including blanched face, single diffuse bar, blanched posterior/caudal region, and erect fins, and combined these into a single “display” category. We observed significant differences among the frequency of behaviors and displays exhibited by Bolbometopon on its spawning grounds (Kruskal-Wallis one way ANOVA, H = 24.36, P < 0.001, n = 9d, Tukey test P < 0.05, Fig. 5). Displays, as well as ascents/descents, were the most commonly observed behavioral interactions taking place during the spawning period. Courtship loop displays, parallel swims, and chases occurred next most frequently, followed by violent head-butts between two males, which occurred least of all. Based on our observations, aggressive interactions in Bolbometopon appear to proceed along the following morphological (color) and behavioral display continuum associated with increasing levels of aggression or stimulation: pale diffuse bar ≤ partial facial blanching (anterior to eye) < ascent/descent ≤ full face blanching (extending posterior to eye) < chase, with full blanching displayed by instigator < parallel swim with both individuals alternately or simultaneously displaying full blanching < head-butt. Numbers of courtship loops were positively correlated with the mean number of fish observed each morning at the aggregation site (Spearman rank correlation, r = 0.83, P = 0.002). In contrast, frequency of displays, ascents/descents, parallel swims, chases, and head-butts were not similarly correlated.

Figure 5 Distribution of behaviors observed at the Bolbometopon muricatum spawning aggregation site at Wake Atoll.

Mean number of overt behaviors and displays (distinct changes in coloration including blanched face, single diffuse bar, blanched posterior/caudal region, and erect fins) observed during the morning spawning period. Different letters above bars represent significant differences with P < 0.05.

Most of the direct observations of spawning that we observed took place in the morning on ebb tides (Fig. 6, Video S5). The spawning we observed on 12 August appears to have taken place on a flooding tide, but this was one of the days our videos were defective and data from the first dive were lost. Low-resolution snippets (single frames) of data recovered from this video show a large and active aggregation of Bolbometopon, and it is likely that additional spawns earlier in the day (during the ebb tide) were lost with the defective video. Personal observations from this first dive of 12 August are consistent with this pattern. Of all the spawns that we observed, that of 23 August took place latest in the day (1300 h), but this was also the only incidence of spawning that we observed outside the aggregation site, approximately 1.2 km to the northwest.

Figure 6 At Wake Atoll, direct observations of spawning Bolbometopon muricatum tended to occur in the morning during ebb tides.

Diamonds indicate days and times where we observed spawning. High and low tides are represented by filled triangles and circles, respectively, and approximate time of sunrise is indicated with a vertical line (Colin, 2010).

We observed spawning or fish assembled at the aggregation site on ten dives where we were also able to geo-reference the dive with GPS track logs. The mean area of these observations was 0.72 ± 0.43 ha. Allowing 0.72 ha to represent the area surveyed while Bolbometopon were present at the aggregation site, we can use the mean number of fish counted on videos of the aggregation site each day to estimate the mean density of Bolbometopon present on aggregation days to be 10.07 fish per ha (Table 1). The single male Bolbometopon seen at the aggregation site from 16 to 19 August during the break in spawning (observed outside the 2 min sampling frame) gives an estimated density of 0.35 fish per ha during that time, an approximately 29 times difference in density from when fish were actively using the aggregation site. A comparison with densities estimated from our more extensive surveys conducted during the break in spawning (including area outside the aggregation site) reveals that Bolbometopon densities could be up to 420 times greater when fish were actively using the aggregation site.

Table 1 Spawning aggregation characteristics of Bolbometopon muricatum at Wake Atoll.

Factor				
	Aggregation days	Non-aggregation days	
N (days)	9	4	
Location surveyed	Aggregation site	Aggregation site	Aggregation site plus outside	
Total area surveyed (ha)	6.48	2.88	359.56	
Density (mean ± SE) per ha	10.07 ± 3.24	0.35 ± 0.35	0.024 ± 0.014*	
Difference in mean density		29	420	
Notes.

* P = 0.007

NOAA CRED towed-diver data from 2005, 2007, 2009, 2011, and 2014 produced 73 separate tows that we evaluated for the densities of Bolbometopon around Wake Atoll. Based on the location around the atoll and the direction of the tow (whether or not the tow crossed the SW aggregation site), eleven tows sampled the aggregation site, while 62 tows sampled other areas of the atoll. CRED surveys of the aggregation site from 2005 to 2014 typically occurred around the full moon (from two days before to six days following) and last quarter moon (four to six days following) and provide additional evidence of a Bolbometopon aggregation during these times. Similar to this study, CRED tows indicated Bolbometopon present at the aggregation site one day before to two days following the full moon, as well as four days following the last quarter moon. Also like the current study, no Bolbometopon were seen at the aggregation site with CRED tows six days following the full moon. Distinct from our work, CRED tows revealed Bolbometopon present at the aggregation site two days before the full moon (before our observations began) and up to six days following the last quarter moon (after our observations had terminated). Mean densities of Bolbometopon across years from the aggregation site tended to be greater (6.2 times) than at other sites around the atoll, but small sample sizes from the aggregation site resulted in a statistical test with low power, and the difference was not significant (9.3 ± 5.2 fish per ha at the aggregation site (n = 11) vs. 1.5 ± 0.28 fish per ha from other sites (n = 62), Mann–Whitney test: T = 457.5, P < 0.427). The maximum density observed at the aggregation site with CRED tows was 51.5 fish per ha compared with a maximum of 10.2 from other sites around the atoll. Comparing morning with afternoon surveys revealed that Bolbometopon densities were greatest (by 9 times compared to other locations) at the aggregation site in the morning, though these differences were not significant (12.6 ± 6.8 fish per ha at the aggregation site during the morning (n = 8) vs. 1.4 ± 0.32 fish per ha from other sites (n = 38)). Surveys of other areas did not show the same temporal differences in densities (1.4 ± 0.32 fish per ha from other sites during the morning (n = 38) vs. 1.7 ± 0.53 fish per ha from other sites during the afternoon (n = 24)). We calculated an island-wide mean (without regard to aggregation site location) of 2.7 (±0.85) fish per ha.

Discussion

During our study at Wake Atoll, we observed giant bumphead parrotfish assembling for spawning in a lek-based system around the full moon, possibly extending to the new moon, in an area encompassing 0.72 ha. Bolbometopon is the largest parrotfish in the world, a highly prized fishery target and cultural resource, and a vulnerable species with ecological and life history traits that have resulted in dramatic population declines throughout much of its range (Dalzell, Adams & Polunin, 1996; Aswani & Hamilton, 2004; Hamilton & Choat, 2012). It is a nearly unique bioeroder and excavator and probable keystone species of Indo-Pacific coral reefs (Bellwood, Hoey & Choat, 2003; Ong & Holland, 2010; Bellwood, Hoey & Hughes, 2012; McCauley et al., 2014), yet until this study, comprehensive descriptions of its reproductive behavior, mating system, and use of spawning sites did not exist. Such details will be key components of conservation efforts and design of marine protected areas (MPAs) for this flagship species (Rhodes & Tupper, 2008; Pillans et al., 2011; Comeros-Raynal et al., 2012; Olds et al., 2014).

Evidence for a lek-like mating system at Wake Atoll

We used a combination of supplemental key videos and photographs to illustrate the full suite of aggressive and reproductive behaviors supporting the designation of a lek-like mating system for Bolbometopon at Wake Atoll. Lek mating systems are not common but are known from a variety of animal taxa including mammals, birds, reptiles, amphibians, insects, and fishes (Emlen & Oring, 1977; Krebs & Davies, 1993; Alcock, 1998). Features of lek mating systems that these taxonomically diverse groups share include the following: (1) the inability of males to economically monopolize either females themselves or resources they require, leading to male defense of territories too small to offer useful resources to visiting females, established solely for the purpose of mating (spawning), and only defended during the spawning period; (2) a great deal of effort by males is spent in territory defense and advertisement to visiting females; (3) territories are often clustered in traditional display areas; (4) females arrive to these display areas solely to select a mate and depart following spawning; (5) mating success is strongly skewed, with males defending certain (usually inner) territories often attaining significantly more matings than males with territories on the periphery; (6) females often outnumber males in the population and travel to the lek site to spawn with the male of their choice; and (7) lek mating systems often correlate with extreme sexual dimorphism (Emlen & Oring, 1977; Moyer & Yogo, 1982; Krebs & Davies, 1993; Alcock, 1998).

At Wake Atoll, Bolbometopon exhibits several of these features associated with lek mating systems, including vigorous defense of small, spatially clustered territories by sexually dimorphic fish, amidst an apparent female-biased population. Due to their size and schooling habit, Bolbometopon should be capable of foraging widely from dispersed resources, which would make resource or female defense economically impossible. The small size of territories defended by males suggests that few resources useful to females could be contained within, although the general clustering of territories during the ebb tide on the down current SW side of the atoll indicates that favorable oceanographic conditions may also play a role in the overall location of the spawning aggregation site. Other features of lek mating systems still require investigation. We do not know from how far females travel to the spawning aggregation site and whether or not they leave the site immediately following spawning. Vigorous aggression and physically costly head-butting suggest a significant benefit for defense of particular territories, although it remains to be determined whether or not centrally located males or territories achieve or are associated with more spawnings than peripheral males/territories. Mating success of individual Bolbometopon may reflect an interaction between oceanographic conditions at the SW side of the atoll, together with an as yet unknown territory or male quality that is preferred by females. Female preference for large size is a possibility, and size is an important determinant of contest success in all phases of sexual selection in a variety of organisms (Warner & Schultz, 1992). Large humphead wrasse (Cheilinus undulatus) males appeared to attain the majority of matings at aggregation sites in Palau, although the synchronous timing of female arrival suggested that small males also achieved some level of mating success (Colin, 2010).

Reproductive aggregations of the phylogenetically related (Westneat & Alfaro, 2005; Cowman, Bellwood & van Herwerden, 2009) C. undulatus (Colin, 2010), the world’s largest wrasse, show a number of similarities with the mating system of Bolbometopon at Wake Atoll. Both mating systems appear to be lek-based and feature early arrival of large males to the aggregation site, establishment of territories that are later visited by females, patrolling by large males high in the water column, and a leisurely spawning ascent by pairs of fish with gamete release in the turbid lagoon plume in the upper 2–3 m of the water column. For both species outside the aggregation period, the aggregation site holds few fish and no resident territorial males are seen. In contrast, no looping, post spawning display, or strong aggression between males were seen for C. undulatus in Palau (Colin, 2010), whereas we observed all of these features at Wake Atoll. Additionally, although the timing of spawning appears to be associated with the outgoing tide for both species, we recorded most observations of spawning in Bolbometopon early in the morning, before noon.

Spawning aggregation behavior during an ebb tide is commonly observed for numerous reef fishes (Choat, 2012; Colin, 2012b; Taylor & Mills, 2013), and most of the spawning that we observed took place high in the water column in the morning on the outgoing tide. Spawning high in the water column on the outgoing tide is believed to advect eggs and larvae short distances away from the reef where predation by benthic planktivores is high (Colin, 2010; Choat, 2012; Colin, 2012b), although shifts in tides may bring the propagules back to the reef (Hamner, Colin & Hamner, 2007). In the Indo-west Pacific, spawning aggregations commonly are found on seaward reef fronts or along the edges of channels, but surveys often report nothing distinctive regarding benthic habitat in the aggregation area, relative to adjacent areas that do not support an aggregation (Colin, 2010; Colin, 2012b). This lack of distinct habitat suggests that use of some spawning aggregation locations may be maintained by social traditions (Warner, 1988) that may be facilitated by the long lifespan of a variety of reef fishes. For example, Bolbometopon is the longest-lived parrotfish (females capable of reaching 40 years, Hamilton & Choat, 2012), and various snappers and surgeonfishes have been determined to reach 50 and 40 years, respectively (Choat, 2012), so ample time and spawning migrations are available for migration routes and aggregation locations to be learned by participants.

Domeier (2012) defined key features of spawning aggregations to include (1) a repeated concentration of conspecific marine animals (2) that is predictable in time and space, and where (3) the density of individuals in the aggregation is at least four times that found outside the aggregation. The spawning aggregation of Bolbometopon at Wake Atoll was repeated and predictable in space and time (morning) over the course of our study and as far back as 2005, where aggregations (of unknown function at the time) of Bolbometopon were observed during biennial surveys by the NOAA CRED (see Fig. 3 in Muñoz et al., 2012). Additionally, the elevated densities (6.2–420 times greater than other locations or non-aggregation days, respectively) of fish and spawning that we observed at the aggregation site bolster Gladstone’s (1986) single observation and indicate that Bolbometopon form true spawning aggregations (sensu Domeier, 2012; Sadovy de Mitcheson & Colin, 2012).

Our study supports the idea that Bolbometopon forms resident spawning aggregations (RAs), with possibly some features (lunar periodicity, migrations) characteristic of transient spawning aggregations (TAs) (Colin, 2012b; Domeier, 2012; Sadovy de Mitcheson & Colin, 2012). Choat (2012) suggested that reproductive aggregations of Bolbometopon (and C. undulatus) showed phylogenetic signal, in that despite their large size (which is associated with TAs) both species are resident spawners. Additionally, the nutritional ecology of species like Bolbometopon, involving nearly continuous feeding activity with high food processing rates, tends to be associated with RAs (Choat, 2012). Resident aggregations are believed to draw individuals to a site within or nearby their adult home range, whereas TAs often involve long distance migrations over many kilometers. We do not currently know the home ranges of adult Bolbometopon at Wake Atoll, but Welsh & Bellwood (2012) recently demonstrated on the Great Barrier Reef (GBR) that the core area of activity for a large excavating parrotfish (the Pacific steephead parrotfish, Chlorurus microrhinos) was restricted to a relatively small area of 1,700 m2. At Wake Atoll, typical feeding groups of Bolbometopon outside the spawning period consisted of 10–20 fish, suggesting that the large numbers that we observed on the aggregation site may have migrated from anywhere around the atoll and merged for spawning (with the aggregation site outside their normal home range). At 11 km long Layang Layang Atoll, only one aggregation of C. undulatus was known for the entire atoll and fish regularly migrated approximately 10 km to reach the aggregation site (Colin & Sadovy de Mitcheson, 2012). Transient aggregations are believed to occur at lunar or seasonal time frames, in contrast to RAs that may occur daily for weeks or months and can occur year round (Colin, 2012b). Studies from the Solomon Islands indicate that Bolbometopon reproduction is strongly influenced by lunar stage (Aswani & Hamilton, 2004; Hamilton, 2004; Hamilton, Adams & Choat, 2008). High gonadosomatic indices for females collected over multiple months from Isabel, New Georgia, and Tetepare Islands suggest that Bolbometopon spawn around the full moon up to the new moon with peak activity in the last quarter (Hamilton, 2004; R Hamilton, pers. comm., 2012), a pattern also seen in a variety of serranids that aggregate to spawn in Melanesia (Hamilton & Kama, 2004; Hamilton, Matawai & Potuku, 2004). Our work also provides support that lunar phase plays a role in Bolbometopon spawning. Based on our estimates of numbers of fish assembled at the aggregation site (decreasing numbers following full moon, increasing numbers following last quarter), we appear to have observed a spawning aggregation that peaked shortly before the full moon, dispersed, then began reforming following the last quarter, and may have continued to the new moon. Additional observations over multiple lunar months would strengthen the importance of lunar phase to Bolbometopon spawning, an unusual pattern for scarine fishes that generally do not show lunar periodicity in spawning (Thresher, 1984; Colin & Bell, 1991; Hamilton, 2004).

Alternative reproductive behavior and density-dependent mating systems

While males at the lek waiting for reproductive schools of females is the most readily observable reproductive behavior at the Wake Atoll aggregation site, less common observations at Wake Atoll, together with direct and indirect observations from Palau and the Solomon Islands suggest that Bolbometopon reproduction may be more complicated. In Palau, presumably where population densities are higher than at Wake Atoll, there is evidence that Bolbometopon aggregate and reproduce in group spawns (Wu, 2013). Density-dependent alternative mating systems are well known in labroid fishes (Warner & Hoffman, 1980; Choat, 2012); for example, males of the circle-cheek wrasse, Halichoeres miniatus, aggressively defend strict harems at low population density sites. In contrast, high density sites are characterized by low levels of male-male aggression, greater female movements, and the potential for female choice of males (McCormick et al., 2010). Given the potential home range size, gregarious habit, and resource distribution of Bolbometopon, harem-based reproduction probably does not occur. Rather, leks may form at low to moderate population densities, with group spawns characteristic of high population densities. At Wake Atoll, where population density of Bolbometopon is low to moderate (island-wide mean of 2.7–2.97 fish per ha), we made no direct observations of spawning interference although we did observe several instances suggesting that sperm competition might occur. The single school of mostly males that we observed suggests that alternative reproductive behavior (Taborsky, 1994; Henson & Warner, 1997) may also be a feature of the mating system. Dominant males may defend territories at the lek and pair spawn with females, while subordinate males may engage in sperm competition and parasitize spawning pairs until they reach sufficient size or competitive ability to acquire a territory (Reinboth, 1973; Robertson & Warner, 1978; Warner & Robertson, 1978; Muñoz & Warner, 2003). Photos from the Palau aggregation (Wu, 2013) as well as histological data from the Solomon Islands (Hamilton, Adams & Choat, 2008) indicate that sperm competition may commonly take place in these locations. Larger reef systems associated with island archipelagos and continental land masses appear to support higher Bolbometopon abundances (where sperm competition may be more prevalent) than isolated oceanic reefs such as Wake Atoll (Hamilton & Choat, 2012).

Gladstone’s (1986) single observation of Bolbometopon spawning from the GBR, and observations from Indonesia and Palau (Hamilton & Choat, 2012; Wu, 2013) show many similarities and some differences with our observations from Wake Atoll. Similarities include positioning of the largest fish at the top of the reproductive aggregation, spawning in the early morning on the ebb tide, full blanching of the face by females moments before gamete release, a leisurely spawning ascent and descent, and gamete release high in the water column between a pair of fish. At Wake Atoll following courtship looping, large males often ascended high into the water column, and ascent/descent was a behavior that we frequently observed from males at the lek awaiting the arrival of females. Ascents/descents and a position high in the water column (or at the top of a school) may be a signal to others in the area, with dominant males potentially spending more time high in the water column. In contrast, Gladstone (1986) did not observe the establishment of temporary spawning territories by males, a dominant feature of the spawning aggregation at Wake Atoll. While it is possible that spawning territories are established on the GBR and these were overlooked by Gladstone’s limited observations, the high abundance of Bolbometopon on the GBR (mean 30 individuals per ha, Hamilton & Choat, 2012) suggests that a lek mating system may not be economically defendable at that location. Additionally, head-butting behavior has never been observed at the GBR despite decades of research (Callaway, 2012). The single pair spawn observed by Gladstone (1986) may have taken place away from the main spawning site and therefore may not have been representative of the reproductive mode of the larger population. Although certain reef fishes are known to display high fidelity to spawning aggregation sites (Zeller, 1998; Claydon, McCormick & Jones, 2012), multiple spawning sites can be active in a particular area; individuals can switch spawning modes (pair spawn vs. group spawn) and are also known to sometimes switch among spawning sites, displaying incomplete fidelity to any given site (Warner & Robertson, 1978; Zeller, 1998; Bijoux et al., 2013).

The un-fished intact population of Bolbometopon at Wake Atoll provides an important comparison to fished sites and intriguingly displays some features consistent (but not conclusive, Sadovy & Shapiro, 1987) with a protogynous sexual pattern: a female biased population, characterized by small females and large males, who may attain disproportionately high reproductive success. Hamilton, Adams & Choat (2008) examined the evidence for protogynous sex change in Bolbometopon and concluded that the populations they examined appeared functionally gonochoristic. This conclusion was based on the lack of transitional individuals observed in their samples, the absence of a female biased sex ratio, and high frequencies of small males occurring around the size that females mature. These authors also indicated it was possible that sampling may have missed transitional individuals and it was unknown whether or not sex-change occurred in any location across the geographical range (Hamilton, Adams & Choat, 2008; Hamilton & Choat, 2012). A size structure with small females and large males can result from protogynous sex change or from differential growth rate between the sexes; male Bolbometopon do display significantly higher growth rates than females (Kobayashi et al., 2011; Hamilton & Choat, 2012). However, lek-based mating systems are characterized by large skews in reproductive success. Although still requiring confirmation, at Wake Atoll large males may better compete for preferred territories, attaining substantially greater reproductive success relative to smaller males and females. Where males gain in reproductive success with size or age faster than females, protogynous sex change is favored (Ghiselin, 1969; Warner, 1975; Munday, Buston & Warner, 2006). Both protogyny and gonochorism have been documented in the parrotfishes, so either life-history pattern is possible for Bolbometopon, although protogyny is much more common in the subfamily. Additional studies at Wake Atoll or from other populations with similar demographic and ecological characteristics may further clarify mechanisms that favor protogyny, reveal transitional individuals in histological samples, and perhaps confirm a protogynous life history. Histological samples from intermediate age and size classes would help resolve the sexual pattern, and given the threatened status of Bolbometopon, such samples might be taken with non-lethal gonadal biopsies (Alam & Nakamura, 2008).

Key components for management consideration

Aspects of the mating system at Wake Atoll will be important to consider for the future management and conservation of Bolbometopon. At Wake Atoll, most of the spawns that we observed took place at the main aggregation site, although we also observed one spawn 1.2 km away, suggesting the possibility of more than one aggregation site. Multiple aggregation sites may decrease the vulnerability of Bolbometopon aggregations to overexploitation, although continued work is necessary before alternate aggregation sites can be confirmed at Wake Atoll and their prevalence determined in other reef systems. At Palmyra Atoll, Pacific steephead parrotfish (C. microrhinos) and bullethead parrotfish (C. spilurus, formerly C. sordidus) also appear to utilize a main aggregation site as well as alternate sites for spawning (R Warner, pers. comm., 2014). On the other hand, the high fidelity of Bolbometopon to the main SW aggregation site suggests that any given spawning aggregation will remain vulnerable once discovered (Zeller, 1998), and it will be important for conservation to determine primary (based on abundance) and alternate aggregation sites. Temporal fidelity to the lunar cycle and spatial fidelity should also facilitate enforcement patrols of seasonal and site-specific closures. The spatial extent of the spawning aggregation at Wake Atoll was remarkably restricted (0.72 ha) given the potential home range utilized by this large-bodied species, and the numbers of fish attending the spawning aggregation compared with foraging schools suggests that smaller schools may merge for reproduction. If true in additional locations, this would indicate the potential for spawning site closures as a conservation measure to supplement other management options such as restrictions to spearfishing while on scuba, or protection of sleeping sites. A number of studies have demonstrated that specific habitat features such as sleeping sites, migration pathways, and reef complexity can be important to the movements of reef fishes and may constrain or facilitate their movements, and relatively small MPAs that incorporate these microhabitats (such as spawning aggregation or sleeping sites for Bolbometopon) can be effective for the protection of a variety of reef fishes, including large parrotfishes (Hamilton, 2004; Meyer, Papastamatiou & Clark, 2010; Hamilton, Potuku & Montambault, 2011; Welsh & Bellwood, 2012; Howard et al., 2013; Taylor & Mills, 2013). The site-attached defense of small territories and larger size of males should make them more vulnerable to exploitation, with potential consequences of such selective harvesting to a variety of life history and ecological traits including operational sex ratio, fertilization success, and intensity of sexual selection (Rowe & Hutchings, 2003; Rowe, Hutchings & Skjaeraasen, 2007; Kendall & Quinn, 2013). Spawning aggregation closures might partially alleviate these issues if fishing outside the aggregations resulted in a more even harvest between the sexes.

Our study of the mating system and spawning aggregation behavior of Bolbometopon muricatum from the protected, relatively pristine population at Wake Atoll provides crucial baselines of population density, sex ratio composition, and productivity of a spawning aggregation site from an oceanic atoll. These baselines will enable assessment of impact or degradation in other, non-pristine locations and provide guidance to managers, conservationists, and stakeholders in terms of needs and objectives for restoration or recovery. Comparisons with larger populations from more extensive reef systems associated with island archipelagos and continental land masses will be important, since the mating and social systems of labroids often show plastic responses to changes in population density, including variation in the timing of sex change in relation to reef-scale structural features (Taylor, 2014). We have demonstrated high fidelity and spatial restriction of a lek-based spawning aggregation to a relatively small area, suggesting that spawning site closures and temporal closures centered around the full to the new moon might form one component of a management and conservation plan for this giant and threatened coral reef fish.

Supplemental Information

Figure S1 A mixed school of male and female Bolbometopon muricatum

Note the male in lower right with a partially blanched face (single arrow), compared with the male in center background displaying fully blanched face, posterior caudal area, and bars (double arrow) apparently chasing one or both males in front of him. The recipients of this aggression can be seen displaying pale bars and partially blanched face.

Click here for additional data file.

Figure S2 Stationary male Bolbometopon muricatum on his territory

During the morning spawning period when stationary male Bolbometopon were not actively engaged in ascent/descent, courtship, or other behaviors, we observed these fish to occupy remarkably small (given their body size) areas of the benthos, as illustrated here.

Click here for additional data file.

Video S1 Large reproductive school of primarily female Bolbometopon muricatum

Only one male fish can be identified in this early morning reproductive school. Also note that very few individuals in this large group of females can be seen feeding.

Click here for additional data file.

Video S2 Ascent/descent advertisement behavior of male Bolbometopon muricatum stationed on their territories within the lek

Two male Bolbometopon can be observed in this video undertaking the ascent/descent display, which also illustrates the spatial distribution and restricted movements of these males during the morning spawning period. Note the foreground male’s fully blanched face at 00:02:11 s and the simultaneous diffuse bar displayed by the two males at 00:28:04 s. The surface of the water is apparent at 00:20:04 s illustrating the height in the water column often attained by males engaged in this display. Note the virtually identical location on the reef utilized by the foreground male as the point of ascent and descent back to the benthos. Depth is 6–7 m.

Click here for additional data file.

Video S3 Courtship looping display of male Bolbometopon muricatum

Shown is a solitary female (lower fish) that is approached and courted in succession by stationary males maintaining position in the water column. Note the barred color pattern displayed by the female (e.g., 00:07:12 s) in response to the male approach, and also that each male returns to its water column station following courtship. This video illustrates courtship of a solitary female for clarity of observation. Frequently, large schools of females move through territories of stationary males and in these cases courtship can occur in succession as described previously, or multiple males will sometimes leave their territories and approach the school and simultaneously court females within.

Click here for additional data file.

Video S4 Chase and parallel swim assessment behavior between three stationary male Bolbometopon muricatum

This video illustrates sequential chases between three males that lead to parallel swims. Note the rapid color changes from diffuse white bar and partially blanched face displayed by recipient males contrasted with full blanched coloration of instigator males. The parallel swims resolve as both males switch direction (00:11:16 s and 00:47:14 s). The diffuse bar displayed by two males in parallel swim (e.g., 00:42:16) fades at the resolution, and all males return to the general location where they were originally located prior to the aggressive interactions.

Click here for additional data file.

Video S5 Spawning sequence of Bolbometopon muricatum

This pair spawn was recorded on 25 August 2011 at 0758 h. Note the temporary monochromatism (including fully blanched face in both male and female) preceding gamete release (at approximately 00:09:01 s). Also note the adjacent male (upper fish) and female (lower) that make no attempt to interfere with the spawning pair.

Click here for additional data file.

We thank J Fleury, T Dotson, J Gehlke, M Spillane, Chugach Support Services Wake Airfield staff, and the 611th ASG, USAF for facilitating the research. J Choat and P Colin provided invaluable advice on locating spawning sites. P Lobel provided logistical advice for working on Wake Atoll. D Meyer, C Price, T Kellison, A Hohn, R Warner, J Choat and M Adreani provided helpful comments on the manuscript. We thank the NOAA Coral Reef Ecosystem Division (National Marine Fisheries Service, Pacific Islands Fisheries Science Center) for providing useful information on the abundance and spatial distribution of fishes at Wake Atoll. The findings and conclusions are those of the authors and do not necessarily represent the views of NOAA.

Additional Information and Declarations

Competing Interests

Author Contributions

Animal Ethics

Field Study Permissions

Roldan C. Muñoz and Bradford Z. Teer are employees of the National Oceanic and Atmospheric Administration. Joseph L. Laughlin is a science and aquaculture consultant at Mariculture Hawaii LLC.

Roldan C. Muñoz conceived and designed the experiments, performed the experiments, analyzed the data, contributed reagents/materials/analysis tools, wrote the paper, prepared figures and/or tables, reviewed drafts of the paper.

Brian J. Zgliczynski conceived and designed the experiments, performed the experiments, contributed reagents/materials/analysis tools, wrote the paper, prepared figures and/or tables, reviewed drafts of the paper.

Bradford Z. Teer and Joseph L. Laughlin performed the experiments, contributed reagents/materials/analysis tools, reviewed drafts of the paper.

The following information was supplied relating to ethical approvals (i.e., approving body and any reference numbers):

All research was conducted in accordance with the Animal Welfare Act (AWA) and with the U.S. Government Principles for the Utilization and Care of Vertebrate Animals Used in Testing, Research, and Training (USGP) OSTP CFR May 20, 1985, Vol. 50, No. 97. The study was conducted on free-living wild animals in their natural habitat and solely involved observations of animals and noninvasive measurements.

The following information was supplied relating to field study approvals (i.e., approving body and any reference numbers):

Permission to conduct field research at Wake Island was granted by Euretha T. Dotson, United States Air Force, Wake Island, HI.

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
