# Peer review of "Spawning aggregation behavior and reproductive ecology of the giant bumphead parrotfish, Bolbometopon muricatum, in a remote marine reserve"

_PeerJ, doi:10.7717/peerj.681_

## Round 0.1 · original submission · Minor Revisions

Overall both referees were enthusiastic about this manuscript and emphasize that it contains essential information relating to the management and conservation of this iconic reef fish. However both also have specific suggestions for improvement of the manuscript prior to - particularly the Discussion, which both referees feel is verbose and the first referee feels should focus specifically on reef fish mating systems rather than lekking in general. I absolutely agree with the reviewers that redundant paragraphs from the results should be cut. Likewise, there are some important omissions identified among the references which should be remedied in revision. Overall, however, these suggestions amount to a careful revision of the discussion and some additional references which is a relatively minor request that should be easily handled by the authors. I look forward to seeing the revised manuscript.

·

Basic reporting

No Comments

Experimental design

No Comments

Validity of the findings

The Discussion needs to be more disciplined with less detail on comparing lek mating systems in a variety of vertebrates and more succinct comparisons with other reef fihes.

Additional comments

Munoz et al Spawning aggregation behavior 1 and reproductive ecology of the giant bumphead parrotfish, Bolbometopon muricatum, in a remote marine reserve
Review.
General
The study represents a valuable summary of reproductive behaviours, lunar and diurnal structure in the timing of spawning and spatial location of spawning sites. As such it is an important contribution to the study of a large vulnerable reef fish the largest of all the scarine labrids Bolbometopon muricatum. However as a general comment I believe the authors have spent too much time in discussing comparative aspects of leking behaviour among very disparate groups of vertebrates, reef fish, mammals, shore birds and surprisingly little time examining difference in reproductive behaviours in ecologically and phylogenetically more similar taxa. For example it is unlikely that the mating system of sandpipers (line 510) will be more informative than the descriptions of sites used by aggregative spawning reef fishes. Descriptions of aggregative spawning sites including predictions concerning what fish may be selecting in terms of reef configuration and oceanography are available in Sadovy and Colin’s book (2012) and also in Colin PL (2010) Journal of Fish Biology 76, 987–1007.
I found omission of the last reference puzzling as Cheilinus undulatus has ecological similarities to B.muricatum in terms of size and is phylogenetically far closer (thanks to Westneat and Alfaro 2005 Phylogenetic Relationships and Evolutionary History of the Reef Fish Family Labridae). Indeed Colin deals with the possibility of lekking behaviour in this species and his descriptions of actual spawning behaviour have a number of instructive parallels with B.muricatum. My recommendation is that this paper be included at the expense of some of the extensive referencing of reproduction in mammals.

A second issue concerns the use of the term IP and TP to describe the different sexes as observed during spawning behaviour (lines 180-193). Differentiation here is based on morphology although males manifest temporary spawning patterns (white face) during reproduction. The terms IP and TP were defined and have subsequently been used in the context of sexual dichromatism Warner & Robertson (1978) Page 3. Extending the use of the terms to cover morphological differences and instances of very temporary spawning color manifestations is simply going to confuse the terminology. It is clear from analysis of sexual phases in the scarine labrids that a large number of species display strikingly different initial and terminal color phases. The presence of primary males bearing the initial phase in a number of species precludes the possibility of making this a straightforward sexual classification. A further complication is that certain ecological groups of scarine have only one color phase. These are seen mainly in large species that feed and forage in schools, Scarus guacamaia and S.coelestinus as indeed is B.muricatum are cases in point. More interestingly some taxa appear to be losing a color phase. In Chlorurus microrhinos in the Pacific Ocean the red phase equivalent to the IP makes up only 1-5% of the reproductive population. The majority of reproductively active females have a green phase shared with the TP. However in the Indian Ocean the sister species the monandric Chlorurus strongylocephalus has a normal size and sex distribution of a red phase IP and a green phase TP. As the analysis of IP and TP phases and the selective factors driving the loss of one of the phases (usually the IP) is likely to get more complex I recommend that the authors do not use these terms. Rather they appear to be morphologically distinct groups. However if they are going to pursue this I would at least be aware of Min Li & Sadovy de Mitchelson (2011) Copeia (2):315-318. Partitioning morphology by sex may not always be straight forward.

As a general comment I found the Discussion very long and convoluted. I estimates about 7 topics were dealt with. For most Discussions 4-5 topics is more appropriate for reader evaluation and understanding of the main messages. As I said above the comparative aspect might be better served by concentrating at this stage on reef fishes.

Specific :

Line 35. Extended age may not be a factor associated with vulnerability to fishing on parrotfishes (Taylor et al 2013 Life histories predict vulnerability to overexploitation in parrotfishes Coral Reefs). For B.muricatum the overwhelming factor is the formation of predictable shallow water sleeping sites.
Line 59-64. They certainly move a lot of sediment and eat of lot of coral but I can’t see this as defining a keystone effect.
Line 134. I suspect the issue in this paper alludes mainly to reef sharks but if towed diver data are used it should be considered. Rizzari, J. R., A. J. Frisch, and S. R. Connolly. 2014. How robust are estimates of coral reef shark depletion? Biological Conservation 176:39–47.

Line 226-229. The ms and references are already very long. I’d leave this sort of observation out of it.
Lines 234-238. Rate of feeding will be influenced both by reproductive activity and time of day. They are confounded here,
Line 258 “temporary dichromatism” which is why it’s not a good idea to start talking about IPs and TPs.
Line 313-341. This would benefit from a BRIEF comparison with C.undulatus as described in Colin (2010).
Line 549-560. It’s almost certain that species dependent on high rates of daily grazing and rapid throughput time of gut material are going to be resident spawners. (See Choat 2012 In Reef Fish Spawning Aggregations: Biology, Research and Management Sadovy & Colin pp 85-116
Line 554 Single obs I’d leave out. Welsh & Bellwood(2012) demonstrate that one species of a large excavating scarine has a very limited home rang.
Lines 634-658. None of these issues are going to be resolved until gonads from intermediate age and size classes are examined. Field observations can only tell you so much.
Line 658. Hamilton and Choat (2012) Fig 12.4 provide a more explicit picture of differences in growth rates.

·

Basic reporting

No comments

Experimental design

No comments

Validity of the findings

No comments

Additional comments

This manuscript is a worthy addition to the growing literature on reproductive ecology of important fishery species. The more we learn about the mating behaviors, the better we will be able to manage them. This is particularly difficult with larger, highly mobile species. The authors have repeated visits to the same sites over several years and are putting together a picture of this species' habitat use and behaviors surrounding spawning. This piece of work is certainly publishable and I suggest only minor revisions to clarify the methods and data and trim down repetitive text.

Here are a few specific comments/suggestions I have for both this manuscript and future work:

-You may want to describe the diver tow methods and explain why you chose to use that method. Many may not understand how it works and what the advantages of such a method are in this type of study.

-How many spawns did you actually observe? While spawning ascent was slow, was the gamete release slow or a quick release right at the surface? More details on the actual spawning observations would be helpful. In addition, more data on what the females are doing during these events would really clarify what is happening leading into and during the spawning event.

-Do you see recruits or other small individuals in the area that are not part of the aggregations? Were any other fish species spawning during that time frame? Did you note any predation on eggs while they were spawning (which may help to explain the timing of spawning)?

-Figure 6 is too busy and it is difficult to see the spawning events among all other lines and dots. I think rearrangement of the graphical presentation would be helpful.

-Is Figure 7 really necessary? These changes in densities are discussed in the text and I don't think it needs to be shown again in graph form.

-The authors already mention that the sexual pattern is not entirely known, but I think it is imperative to know this in order to make any suggestions about future management of this species. Is it possible to either sacrifice a small number or to take gonad plugs (non-lethal), as has been done with Goliath grouper?

-The discussion section could really be trimmed down and made much more concise. There are paragraphs that are repeated from the results and don't really add anything to the main points of how these data can be helpful in conservation efforts.

---

## Round 0.2 · accepted · Accept

After reading through your revised manuscript I see no reason to return it to the referees, both of whom were supportive of publication following minor revisions. I am satisfied that your revisions have addressed the referee comments and thank you for your careful attention to the referee comments. Overall, I feel this work is an important and welcome contribution to the field, so I am happy to accept your manuscript to move forward into press.